# Correlated Responses to Selection for Intramuscular Fat on the Gut Microbiome in Rabbits

**DOI:** 10.3390/ani14142078

**Published:** 2024-07-16

**Authors:** Marina Martínez-Álvaro, Agostina Zubiri-Gaitán, Pilar Hernández, Cristina Casto-Rebollo, Noelia Ibáñez-Escriche, Maria Antonia Santacreu, Alejandro Artacho, Vicente Pérez-Brocal, Agustín Blasco

**Affiliations:** 1Institute for Animal Science and Technology, Universitat Politècnica de València, 46022 Valencia, Spain; 2Area of Genomics and Health, Foundation for the Promotion of Sanitary and Biomedical Research of Valencia Region (FISABIO-Public Health), 46022 Valencia, Spain; 3Biomedical Research Networking Center for Epidemiology and Public Health (CIBERESP), 28029 Madrid, Spain

**Keywords:** genetic selection, intramuscular fat, gut microbiome, correlated responses

## Abstract

**Simple Summary:**

In meat production, the fat within an animal’s muscles, known as intramuscular fat (IMF), is key to quality. This study investigated how breeding rabbits for higher or lower IMF levels affect their gut microorganisms. We focused on a rabbit population that was bred over 10 generations to achieve either high-IMF or low-IMF levels. Our findings show that genetic selection changes the gut’s microbial composition, with more noticeable differences at the genus level than at the broader phylum level. High-IMF rabbits had different abundances of *Escherichia*, *Methanobrevibacter*, and *Hungateiclostridium* microorganisms compared to low-IMF rabbits, amongst others. We identified four specific microorganisms that could predict a rabbit’s IMF genetic line with 78% accuracy. This research highlights the link between muscle fat genetics and gut microorganisms, opening the possibility of developing microbiome modulation strategies to influence IMF in animals, which could improve meat quality.

**Abstract:**

Intramuscular fat (IMF) content is important for meat production and human health, where the host genetics and its microbiome greatly contribute to its variation. The aim of this study is to describe the consequences of the genetic modification of IMF by selecting the taxonomic composition of the microbiome, using rabbits from the 10th generation of a divergent selection experiment for IMF (high (H) and low (L) lines differ by 3.8 standard deviations). The selection altered the composition of the gut microbiota. Correlated responses were better distinguished at the genus level (51 genera) than at the phylum level (10 phyla). The H-line was enriched in *Hungateiclostridium*, *Limosilactobacillus*, *Legionella*, *Lysinibacillus*, *Phorphyromonas*, *Methanosphaera*, *Desulfovibrio*, and *Akkermansia*, while the L-line was enriched in *Escherichia*, *Methanobrevibacter*, *Fonticella*, *Candidatus Amulumruptor*, *Methanobrevibacter*, *Exiguobacterium*, *Flintibacter*, and *Coprococcus*, among other genera with smaller line differences. A microbial biomarker generated from the abundance of four of these genera classified the lines with 78% accuracy in a logit regression. Our results demonstrate different gut microbiome compositions in hosts with divergent IMF genotypes. Furthermore, we provide a microbial biomarker to be used as an indicator of hosts genetically predisposed to accumulate muscle lipids, which opens up the opportunity for research to develop probiotics or microbiome-based breeding strategies targeting IMF.

## 1. Introduction

Intramuscular fat (IMF) content is an important trait with a dual perspective. In farm production, a high-IMF content is a main objective of the meat industry, as it increases the quality of meat in terms of sensory attributes such as juiciness, tenderness, and flavor [1]. In terms of human health, increased deposition of IMF, along with increased lipid deposition in other fat depots, contributes to obesity [2]. This is a major public health and economic concern because obesity prevalence has markedly increased worldwide over the last 40 years, and it is projected to keep increasing in the coming years [3]. Intramuscular fat is a multifactorial trait to which dietary interventions and host genetic factors are major contributors, with heritabilities ranging from 0.40 to 0.70 in farm animals [4,5], resembling those reported for general obesity in humans [6]. In recent decades, attention has also focused on the microbiota and its role in fat accumulation, as there are a variety of microbial mechanisms that influence host energy management by altering appetite [7], energy production from food intake [8,9,10,11,12,13,14], or inflammatory and immune processes through increased production of microbial metabolites [9,12,13,14,15]. Dietary interventions and host genetic factors also influence microbiota composition [8,9,10,11,12,13,14,16]. For example, the abundance of microbial genera *Akkermansia*, which affects obesity, is associated with a marker near PLD1, a gene already known to influence body mass index [17]. Since host genetics, the environment, and the microbiome form a complex biological system that influences lipid metabolism, it is not easy to disentangle the individual effects of any of these factors on IMF and, more importantly, to determine a direction of causality [18]. In addition, there are differences between experiments in terms of design, sample collection and DNA treatment, bioinformatic procedures, statistical methods, data transformations, and/or failure to account for lifestyle or external factors. As a result, some outcomes remain inconclusive; for example, although a relationship between the ratio of Firmicutes to Bacteroidetes (FB ratio) and obesity was originally proposed [19,20,21,22,23], many inconsistencies found later on have yet to be investigated [24].

One practical way to respond to these controversies is to isolate individual factors in order to provide answers to specific questions using appropriate experimental designs. This is not always possible, especially in humans, where there are many confounding factors, but it is easier in farm or laboratory animals where controlled conditions are easier to achieve. At the Universitat Politècnica de València, a 10-generation divergent selection experiment for IMF in the Longissimus muscle was performed in rabbits [4,25], with a response of 0.49 g/100 g, equivalent to 47% of the mean of the trait or 3.8 standard deviations (SD) at the end of the experiment [26]. The divergent genetic lines (high-IMF or H-line and low-IMF or L-line) were raised at the same time, kept in the same environment and with the same management, and fed the same diet throughout the experiment. The differences observed in their microbiome can be directly attributed to the genes underlying IMF and genetically correlated traits, independent of any genetic parameters or models, constituting the main advantage of our experimental design, compared with other microbiome-genetic studies. The results of this experiment will have an influence on the production of rabbit meat, mainly consumed in China, the Democratic People’s Republic of Korea, Egypt, Spain, and France [27], but they will also serve as a model for other farm species in which IMF is an important meat quality factor. On the other hand, the results of this experiment can be used for orientating the studies of lipid metabolism in humans, because a rabbit’s lipoprotein metabolism, cardiovascular system, and clinical symptoms associated with obesity are more similar to those in humans than the same systems in mice [28,29]. In previous studies, we have shown that the selection for IMF in the Longissimus modified the fat content of various muscles [30] and other body fat depots [4], increased the lipogenic activity of muscle, fat [31], and liver tissues [32], affected the liver weight [32], and influenced the adipocyte size [31], with all levels greater in the H-line than in the L-line, impacting the muscle and liver fatty acid profile [26,33]. With the -omics revolution, these lines became extremely valuable experimental material to decipher which modifications in -omics variables are regulated by IMF genetics. In a genomic study, we identified several genes related to the IMF content and its fatty acid composition, which were modified by the selection [34,35]. In a plasma metabolomic study, we found an impaired ability of the L-line for β-oxidation of fatty acids, indicating a limited capacity to obtain energy from lipids, as well as alterations in plasma levels of metabolites derived from the microbiome, including catabolites from branched-chain and aromatic amino acids, and secondary bile acids [36]. In a functional metagenomics study [37], we observed that microbial biosynthesis of lipopolysaccharides, peptidoglycans, lipoproteins, mucin components, and bile acids, among others, are influenced by host genetic determination of lipid accumulation in muscle, which was consistent with some of the results obtained in the plasma metabolomics study mentioned above. However, many questions remained regarding the taxonomic composition of the cecum microbiome. For example, some microbial genes involved in mucin degradation or methanogenesis were altered by the selection, but it was unclear whether the selection changed the abundance of the major mucin degrader *Akkermansia* or the methane producer *Methanobrevibacter*. It was also not clear whether the selection altered the FB ratio, a major indicator of obesity, showing if it has a hereditary component. The study of the gut taxonomic composition of these lines opens the possibility of defining microbiome changes driven by host genetics for IMF and to define which are the main microbial organisms involved in their functional differentiation. 

This work aims to investigate the correlated response in gut taxonomic microbiota composition following a 10-generation divergent selection experiment for IMF in rabbits under controlled environmental conditions using whole-metagenome sequencing of 89 cecal content samples. The output of this research will clarify whether specific taxa can be associated with IMF-divergent genetic lines and will allow the identification of specific microbial biomarkers indicative of the host genetic predisposition to accumulate lipids in muscle.

## 2. Materials and Methods

### 2.1. Animal Material and Sampling

All experimental procedures involving animals were approved by the Research Ethics Committee of the Universitat Politècnica de València, according to Council Directives 98/58/EC and 2010/63/EU (reference number 2017/VSC/PEA/00212). 

The rabbits used in this experiment belong to two lines divergently selected for IMF in the Longissimus muscle. The divergent selection experiment was performed at the Universitat Politècnica de València and is widely explained in works by Zomeño et al. (2013) [38] and Martínez-Álvaro et al. (2016) [4]. The H-line and L-line were created by selecting for higher or lower IMF content, respectively. The selection criterion was the average IMF value measured at nine weeks of age on two full siblings from each doe’s first parity. Each line was composed of 10 males and 60 females; the lines were contemporarily reared and selected under the same environmental conditions, including the same amount and type of commercial diet, to ensure that the same environmental factors affected both lines. The experiment was successful, yielding a direct response to selection in the 10th generation of 0.49 g/100 g of muscle [26], equivalent to 47% of the mean and 3.8 SD of the trait. 

This study was performed with 47 rabbits from the H-line (27 males and 20 females) and 42 from the L-line (22 males and 20 females) from the 10th generation of selection, all randomly chosen. Animals came from 44 different litters of 44 different does (21 from H and 23 from L) with an average of 2 animals sampled within each litter (SD of 0.9 and range from 1 to 5), a male and a female. The animals were weaned at four weeks of age and were then reared jointly within litters until slaughter at nine weeks of age. During the fattening period (from weaning to slaughter), the animals were fed ad libitum with the same commercial diet, the composition of which is described in Appendix A. The animals were deprived of food four hours before slaughter. The slaughter was performed by exsanguination after electrical stunning; the gastrointestinal tract was removed from the abdominal cavity, and the cecum content was collected in 50 mL sterile Falcon tubes, homogenized, and aliquoted in 2 mL cryogenic tubes. The aliquots were immediately submerged in liquid nitrogen and then stored at −80 °C until analysis. The carcasses were then chilled for 24 h at 4 °C, after which the Longissimus muscle was excised, minced, and freeze-dried. The IMF content was finally quantified as g/100 g of fresh muscle using near-infrared spectroscopy, applying the equations developed by Zomeño et al. (2011) [39].

### 2.2. Microbial Abundance Measurements

The isolation and quality assessment of bacterial DNA were conducted following the methods described by Martínez-Álvaro et al. (2021) [37]. Similarly, the sequencing and library preparation procedures were performed in two different laboratories and were thoroughly detailed in the same study [37], as they were performed using the same samples. The raw FASTQ files obtained were initially filtered and trimmed using Trimmomatic v0.39 [40], to obtain high-quality reads, setting the following parameters: leading: eight (removal of nucleotides at the 3′ end when quality < eight), trailing: eight (removal of nucleotides at the 5′ end when quality < eight), slidingwindow: 10:15 (scan the read with a 10-base sliding window, cutting when the average quality drops below 15), and minlen: 50 (discard reads with less than 50 bases). After the quality filter, contaminant reads from the host were removed by mapping the reads against the reference rabbit genome (OryCun v2.0.101) using the Bowtie2 software v4.1.2 [41], and removing the mapped reads using the SAMtools software v1.2.1 [42]. 

The clean FASTQ files obtained were processed with the SqueezeMeta fully automatic metagenomics analysis pipeline [43], using the co-assembly mode with default parameters. Briefly, the pipeline uses Megahit [44] to assemble the reads into contigs, which are then filtered by length using prinseq [45] (minimum 200 bases long). Gene prediction is then carried out on the contigs using Prodigal [46], followed by a homology search against the GenBank nr database using Diamond [47] for their posterior taxonomic assignment. A last common ancestor (LCA) algorithm, implemented using custom scripts developed by the authors [43], is used to assign the corresponding taxonomy to the predicted genes and, finally, to the contigs. To estimate the final abundance dataset, the reads of each sample are mapped back against the contigs using Bowtie2 [41], and the extraction of the raw number of reads mapping to each contig is performed using BEDTools [48]. Further details on the bioinformatics pipeline can be found in the work by Tamames and Puentes-Sánchez (2019) [43]. The resulting dataset was split into the corresponding taxonomic ranks (phylum, class, order, family, genus, and species), and only phylum (*n* = 173) and genus (*n* = 1229) were analyzed in this study.

### 2.3. Effect of Selection on Alpha Diversity

The diversity within samples (i.e., α-diversity) was calculated at the phylum and genus levels with data in raw counts. The adjusted Shannon index (${H^{\prime }}_{adj}$) was used, which is the Shannon index ($H^{\prime }$) expressed as a proportion of the maximum value of $H^{\prime }$ that would be found if all phyla or genera had the same abundance (${H^{\prime }}_{max}$) [49]. The Shannon index is defined as follows: (1)$$H^{\prime }=\sum_{1}^{S}-[\rho_{j}\ln\left(\rho_{j}\right)]$$
where S is the total number of phyla or genera found in a sample, and $\rho_{j}$ is the j-th phylum or genus expressed in relative abundance. Then, ${H^{\prime }}_{max}$ is the Shannon index calculated when $\rho_{j}=1/S$. Finally, the adjusted Shannon index for each sample was calculated as follows:(2)$${H^{\prime }}_{\mathrm{adj}}=\frac{H^{\prime }}{{H^{\prime }}_{max}}$$

The effect of selection for IMF on the alpha diversity was computed by comparing the adjusted Shannon index distribution of the lines by fitting a Kruskal–Wallis test, after correcting the values for a sequencer effect.

### 2.4. Effect of Selection on the Microbial Abundance

#### 2.4.1. Microbial Abundances Transformations

Before analyzing the correlated responses to selection on microbiome composition at the phylum and genus levels, we discarded any phylum or genus that was undetected in more than 10% of the samples. Doing so resulted in the retention of 63 phyla and 315 genera, accounting for over 99.8% of the original counts. The remaining missing phyla and genera were imputed using a Bayesian-multiplicative method [50]. For a raw description of the cecum microbiota composition, the relative abundances of microbial taxa were computed. For subsequent analyses, we transformed microbial abundances (in raw counts) using the additive log–ratio (alr) method to address their compositional nature and prevent misleading conclusions [51]. The reference taxa (phylum or genus) used as the denominator was selected according to the criteria established by Greenacre et al. (2021) [52]. Fibrobacteres (for the phylum dataset) and *Eubacterium* (for the genus dataset) were the reference taxa finally selected, as discussed later in the Results section. 

The possible effect derived from sequencing samples in two different laboratories was evaluated by a principal component analysis (PCA) of the alr-transformed microbiota composition (Appendix A). A clear sequencer machine effect was observed, which was successfully removed after correction for a sequencing effect as follows:(3)$$x_{j}={Seq}_{j}+e,\;\;j=1,\ldots ,J$$
where $x_{j}$ is the alr-transformed abundance of the j-th taxa and ${Seq}_{j}$ is the sequencer effect of the j-th taxa. All further analyses were performed with corrected data.

#### 2.4.2. Microbiability of IMF

The proportion of the variance of *IMF* attributable to microbiota composition variance (at phylum and genus level) was estimated using a Bayesian analysis [53] on the following MBLUP model [54]: (4)IMF=Xb+Wm+e

Data were assumed to be normally distributed as follows:(5)$$IMF|b,m,\sigma_{e}^{2}\sim N(Xb+Wm,I\sigma_{e}^{2})$$
where IMF is the vector with the phenotypes of the traits, b is a vector that includes sex as a fixed effect, m is the random microbiome effect, e is the vector of residuals and X and W are the known incidence matrices for fixed and microbiome effects. Microbial random effects were assumed to be normally distributed with mean 0 and variance equal to ${K\sigma }_{m}^{2}$, $\sigma_{m}^{2}$ is the microbiome variance, K is the microbiome relationship matrix between individuals, computed as $\left(\frac{1}{p}\right)BB^{T}$, B is a matrix containing the additive log-transformed data (standardized) with dimension n × p, where n is the number of individuals and p is the number of alr-transformed phyla or alr-transformed genera. Residuals were assumed to be normally distributed with a mean of 0 and a variance equal to ${I\sigma }_{e}^{2}$, where $\sigma_{e}^{2}$ is the residual variance. Microbiome and residuals were assumed to be independent. The analyses were carried out using BGLR Bayesian software [55]. Bounded flat priors were used for fixed effects and variances. Marginal posterior distributions of the estimated parameters were based on Markov chain Monte Carlo (MCMC) methods, consisting of 120,000 iterations, with a burn-in period of 20,000, and only 1 of every 10 samples saved for inferences. Convergence was tested with R package coda [56] by checking the Z criterion of Geweke. Microbiability was computed for each MCMC sample as the ratio between $\sigma_{m}^{2}$ and $\sigma_{m}^{2}+\sigma_{e}^{2}$, and the median of its marginal posterior distribution, the highest posterior density interval at 95% probability, and the probability of the microbiability of being greater than 25% was calculated. Additionally, microbiability was computed within the line, fitting the MBLUP model with animals from the H-line or the L-line.

#### 2.4.3. Correlated Response to Selection for IMF on the Microbial Abundances

The study of the gut microbial taxa abundances modified after selection was performed by partial least squares-discriminant analysis (PLS-DA) and linear partial least squares (PLS). The dependent variable of the PLS-DA was a categorical vector coding the H-line or L-line, while that of the PLS was a vector containing the IMF content. After standardization (centering and scaling) of the datasets, all analyses were performed using the R package mdatools [57]. The adjustment and validation of the multivariate models were performed using a two-step cross-model validation procedure (CMV). The first step focused on selecting microbial taxa (genus or phylum) based on whether the confidence interval of their regression coefficients excluded 0, and on their “variable importance in the projection” (VIP) value. This was done under an eight cross-validation procedure to determine the cut-off value for VIP (between 0.8 and 1.1), repeating the process 20 times and fitting a total of 160 models. The taxa selected in more than 70% of the models were considered in the second step. The second step aimed to evaluate the performance of a final model that included the selected taxa. Again, an eight-fold CV strategy repeated 20 times was used, fitting a total of 160 models. The performance of each of the final models was evaluated using a misclassification table for the PLS-DA, and the Q^2^ parameter for the PLS, and the average and SD were reported. Additionally, a permutation test was performed within each final model, where the H-line or L-line for the PLS-DA, or IMF content for the PLS, was permuted. The CMV procedure is described in more detail by Zubiri-Gaitán et al. (2023) [36].

Microbial taxa that fit both the final PLS-DA and PLS models were considered to have a correlated response to selection for IMF. The correlated response was quantified by fitting a linear model per microbial taxa, including its alr-transformed abundance as a dependent variable, and line and sex as fixed effects. Bayesian inference was used with flat priors for all unknowns. Analyses were run with the R package RabbitR (https://github.com/marinamartinezalvaro/RabbitR, accessed on 1 April 2024). After some exploratory analyses, results were based on marginal posterior distributions of 60,000 iterations, with a burn-in period of 10,000, and only one of every 10 samples was saved for inferences. In all analyses, convergence was tested using the Z criterion of Geweke. The parameters obtained from the marginal posterior distributions of the differences between lines were the mean; the highest posterior density region at 95% probability (HPD95%); and the probability of the difference being greater than 0 when the mean was positive or lower than 0 when the mean was negative (P_0_). Differences were expressed as units of SD of the microbial taxa abundances after pre-correcting them for line and sex.

### 2.5. Search of Microbial Biomarkers for Prediction of IMF

While the PLS-DA model can identify variables able to classify the lines, it employs a multivariate approach that includes all microbial variables contributing, even if some contain redundant information. However, from a practical point of view, it might be interesting to find a single microbial biomarker that is composed of the smallest number of microbial taxa and has the highest classification accuracy between lines. To find this microbial biomarker, a procedure similar to that used in stepwise forward regression was used, but instead of adding the microbial taxa as simple covariates, they are added as part of what—in compositional data analysis—is called a balance or a particular isometric log–ratio, i.e., part of a log contrast in which the numerator and denominator are geometric means of multiple raw microbial abundances (see Rivera-Pinto et al. (2018) [58] for a full description of the method). The R package selbal [58] was used to design such a microbial biomarker using the 51 microbial genera that had a correlated response to selection. The algorithm takes into account the compositional nature of the microbial data and requires that the data be entered in relative abundances. In the first step, the number and selection of variables composing the balance were based on the least classification error using the function selbal.cv as recommended by Rivera-Pinto et al. (2018) [58], where the k-fold was equal to eight and the number of iterations was equal to 20. In a second step, the selected balance was fitted in a logit regression model to test its classification ability between lines, in a new eight-fold CV, repeated 20 times with the selbal function. The dataset was divided into eight groups; 7/8 of the datasets were used to fit a logit regression using the balance as a covariate; and the classification ability of the model was evaluated based on the test group set aside. The average classification accuracy obtained across all folds and replicates was reported. In addition, the entire procedure was repeated limiting the number of microbial genera making up the balance to five.

## 3. Results

### 3.1. Cecum Microbiome Composition in Rabbits

The cecal microbiome was mainly composed of Firmicutes (83.12% of the total abundance) and Bacteroidetes (13.45%) phyla, and other subdominant phyla such as Verrucomicrobia (1.19%), Proteobacteria (0.76%), Actinobacteria (0.76%), and Tenericutes (0.1%). Other bacterial phyla such as Fibrobacteres, Chloroflexi, Armatimonadetes, and Elusimicrobia were less represented with relative abundances ≤ 0.01%. Methanogenic archaea from the phylum Euryarchaeota had a relative abundance of 0.04%, while the fungal phylum Ascomycota accounted for 0.12%, Chytridiomycota for 0.002%, and Basidiomycota for 0.0001%. At the genera level, *Ruminoccocus* was the most abundant identified genus (21.7%), followed by *Alistipes* (19.6%), *Bacteroides* (5.43%), *Clostridium* (5.16%), *Faecalibaculum* (4.45%), and *Akkermansia* (3.54%). The most abundant methanogenic archaea was *Methanobrevibacter* with 0.52% abundance (see Appendix A). 

Because microbiome data are compositional, log–ratio transformation is strongly recommended to avoid spurious results when comparing groups or calculating microbiome-microbiome or microbiome-trait associations [51]. Recently, we proposed analyzing –omics data using an additive log–ratio transformation in which the denominator is a variable with low variance between samples and high Procrustes correlation with the standard pairwise log–ratio system [52], so it does not bias the true distances between samples. In the taxonomic composition of the cecal microbiome, phylum Fibrobacteres, and genus *Eubacterium* were the best candidates for use as denominators in the transformation because they preserved the Euclidean distances between individuals calculated based on all the pairwise log–ratios standard (Procrustes correlations of 0.992 for phylum and 0.998 for the genus, see Appendix A). The variances of their natural logarithms were very small (0.0165 for phylum and 0.0442 for genus), which attributes most of the variation in the additive log–ratio to the numerator, simplifying its interpretation.

### 3.2. Microbiability of IMF

The microbiability of IMF was 44.5% [15.4%, 74.8%] when computed based on microbial genera, with a probability greater than 25% of 0.86. When computed based on phyla abundances, it was lower, 23.7% [7.4%, 43.8%], with a probability of being greater than 25% of 0.38. We also computed microbiability based on genera calculated separately for each line, and it was still considerable, 38.8% [12.9%, 70.9%] in the H-line (probability of being greater than 25% was 0.81) and 43.2% [13.5%, 76.2%] in the L-line (probability of being greater than 25% was 0.83).

### 3.3. Divergences between IMF Lines in the Gut Microbiome as a Response to Selection

Alpha diversity was not associated with host IMF genes, as the H-line and L-line had similar microbial diversity, measured as an adjusted alpha diversity index at either the phylum (${H^{\prime }}_{adj}=$ 0.13 with s.e. of 0.02 for the H-line, and ${H^{\prime }}_{adj}=$ 0.12 with s.e. of 0.02 for the L-line, a *p*-value of the Kruskal–Wallis test was 0.29) or genus level (${H^{\prime }}_{adj}=$ 0.51 with s.e. of 0.04 for the H-line, and ${H^{\prime }}_{adj}=$ 0.51 with s.e. of 0.037 for the L-line, *p*-value = 0.85). 

However, selection altered specific microbiome abundances in the cecum at both the phylum and genus levels, with the effects being more pronounced at the lower taxonomic resolution. Based on 14 alr-transformed phyla and 66 alr-transformed genera, PLS-DA models presented a cross-validation classification accuracy between lines of 71.0% (phyla) and 94.3% (genera), which decreased to 51.8% (phyla) and 48.9% (genera) when permuting the labels of the samples (Appendix A). In addition, 17 alr-transformed phyla and 74 alr-transformed genera identified predicted IMF variation with a Q^2^ accuracy of 11.8% (phyla) and 65.7% (genera), in cross-validation (Appendix A), which decreased to −14% (phyla) and −31% (genera), when IMF values were permuted. Moreover, 10 alr-transformed phyla and 51 alr-transformed genera overlapped in both PLS-DA and PLS models and were assumed to show a correlated response to selection for IMF (Appendix A and Figure 1a,b). 

Correlated responses to selection on the 10 and 51 alr-transformed microbial phyla and genera abundances were quantified as the differences between lines (in units of SD, also known as effect size) by fitting a linear model that included the sex effect (Figure 1c and Appendix A). At the phyla level, the two lines were dominated by either Verrucomicrobia and Chloroflexi phyla in the H-line (+0.63 SD and +0.42 SD greater abundance, respectively, with a probability of this difference being greater than 0 (P_0_) ≥ 0.97), or Armatimonadetes and Euryarchaeota in the L-line (−0.46 and −0.43 SD, respectively, P_0_ ≥ 0.98). In addition, the H-line was enriched in Proteobacteria (+0.34, P_0_ =0.94), Candidatus Saccharibacteria (+0.33 SD, P_0_ = 0.93), and Elusimicrobia (+0.31 SD, P_0_ = 0.92) bacterial phyla. Interestingly, the two fungal phyla Chytridiomycota and the much less abundant Basidiomycota were also enriched in the H-line (+0.34 SD, P_0_ = 0.94); however, the L-line was enriched in Ascomycota (−0.45 SD, P_0_ = 0.98).

We did not find differences between lines in the FB ratio, which was 7.51 (SD = 0.55) in the H-line and 7.16 (SD = 0.59) in the L-line, and there was low statistical evidence of a greater ratio in the H-line (H-L = 0.38 and SD 0.79, P_0_ = 0.68). However, out of the 51 genera distinguishing the lines, 31 belonged to Firmicutes and five to Bacteroidetes, with enrichment either in the H-line (16 Firmicutes and three Bacteroidetes) or in the L-line (15 Firmicutes and two Bacteroidetes). Among those genera enriched in the H-line, *Hungateiclostridium*, *Limosilactobacillus*, *Lysinibacillus, Porphyromonas*, *Erysipelatoclostridium*, and *Gemminger* showed the largest effect sizes from +0.45 SD to +0.67 SD (P_0_ ≥ 0.98). Among those, Firmicutes and Bacteroidetes genera enriched in the L-line, *Fonticella*, *Megasphaera*, *Exiguobacterium*, *Flintibacter*, *Coprococcus*, *Candidatus Amulumruptor* and *Odoribacter* showed the largest effect sizes from −0.33 SD to −0.61 SD (P_0_ ≥ 0.93). The genera *Ruminococcus* (21.7% of total relative abundance), *Lactobacillus*, and *Turicibacter* were also enriched in the L-line, with effect sizes of −0.39, −0.38, and −0.35 SD, respectively (P_0_ ≥ 0.95), and VIP ≥ 0.86. 

Several Proteobacteria genera also played an important role in differentiating the two lines. The H-line was enriched in *Oxalobacter*, *Desulfovibrio*, *Bartonella*, *Legionella*, and *Sutterella* with effect sizes ranging from +0.37 to +0.57 SD, and P_0_ ≥ 0.96; and the L-line was enriched in *Escherichia* with an effect size of −0.69 SD (P_0_ = 1.00) (Figure 1c). Other interesting results included the enrichment of the abundant genus *Akkermansia* (3.53% relative abundance) in the H-line (+0.31 SD, P_0_ = 0.93), and the enrichment of the ubiquitous methanogenic archaea *Methanobrevibacter* (0.52% relative abundance) in the L-line, with −0.51 SD, P_0_ = 0.99.

### 3.4. Microbial Biomarkers to Predict Host IMF Genetic Background

Finding a simple microbial biomarker that can be used to classify new animals into the two lines could be an attractive outcome to apply in the health or meat quality industry. To this aim, we identified a microbial balance formed by 13 microbial genera abundances, selected amongst the 51 genera having a response to selection. It contained the geometric mean of *Escherichia*, *Megasphaera*, *Lactobacillus*, *Massilistercora*, and *Robinsoniella* in the numerator; and *Legionella*, *Lysinbacillus*, *Hungateiclostridium*, *Methanosphaera*, *Moorella*, *Faecalitalea*, *Phocaeicola*, and *Sutterella* in the denominator (Figure 2a). When this balance was fitted in a logit regression for classification between lines, cross-validated classification accuracy was 91.1% (SD of 1.01%). A less complex balance was additionally obtained by limiting the number of genera comprising the balance to a maximum of five. In this case, a balance formed by four genera composed of the geometric mean of *Escherichia* and *Massilistercora* in the numerator and *Legionella* and *Lysinbacillus* in the denominator (Figure 2b) was selected. This balance was able to classify the lines with an accuracy of 78.2% (SD = 0.4%). 

## 4. Discussion

In this study, we first estimated IMF microbiability (first used by Difford et al. (2016) [59]), which quantifies the percentage of IMF phenotypic variation between individuals due to the variation in their microbiome composition, independent of whether these microbial abundances are influenced by host genetics. IMF microbiability at the genera level was around 45% greater than the estimate at the phyla level. The better fitness observed at lower taxonomic resolution can be explained by the fact that clustering the counts at higher taxonomic ranks could hide the variation among individuals that arises from genera with versatile functions for lipid metabolism. The greater explanatory power at lower taxonomic levels was also observed for resilience traits in rabbits [60]. Although there are no previous records of IMF microbiability in rabbits, other studies performed in pigs provided lower values ranging from 6% [61] to 26% [62,63]. Our higher microbiability could not be attributed to the process of divergent selection since the median of the microbiability distribution estimated using genera calculated separately for each line was still considerable (39% in H and 43% in L). Instead, the differences with other results may be attributed to additional random effects in the models of these experiments, or different methods used to calculate metagenomic distances between individuals [63].

Due to our divergent selection experimental design, we can go a step further and disentangle which of the microbial features related to IMF are influenced by host genetics by studying the differences between the microbiomes of the IMF lines. While vertical transmission of microbes unaffected by host genetics may occur from the selected mother to the kit (for example via the birth canal, lactation, and/or coprophagy of the mother’s soft feces in the nest [64]), the stability of such microbial populations in the digestive tract of the rabbit under standardized environmental conditions remains unclear [65,66,67,68]. In addition, the microbial abundances influencing IMF and only related to the environment should be randomly distributed among the animals of both lines, as the animals were randomly selected for this microbiome study. Considering these factors, we expect that the differences in microbiome features between the IMF lines are primarily due to divergence in host genetics. In addition to genetic changes due to selection, our finite rabbit population is also subjected to random genetic changes due to genetic drift that could also affect the microbiome. This possible genetic drift was alleviated by focusing on the microbial taxa that differed between lines (PLS-DA) and was also linearly associated with IMF (PLS), as first proposed in Martínez-Álvaro et al. (2021) [37]. 

Selection for IMF presented a correlated response in the microbiome composition. Although we did not observe a correlated response to selection for IMF on FB ratio, we observed a correlated response to selection on the abundance of specific genera belonging to these phyla. For example, *Gemminger* and *Erysipelatoclostridium* genera classified within Firmicutes and *Porphyromonas* classified within Bacteroidetes were enriched in the H-line. In the literature, *Gemminger* was found to be more abundant in obese children than in normal-weight individuals [69]; *Erysipelatoclostridium*, was linked to diet-induced obesity in gnotobiotic mice [70]; and periodontal infection with *Porphyromonas* induced obesity, glucose tolerance, hepatic steatosis, and altered endocrine function in mice [71,72]. In this study, we suggest these associations have a host-genetic component.

Another advantage of this experimental material is that its genomic, metabolomic, and cecum functional profile has been described in detail in previous studies [34,35,36,37], so that the results of this study can be integrated with other -omics. For example, the L-line presented a greater abundance of *Flintibacter*, linked to bile acid metabolism [73], and of *Escherichia* and *Lactobacillus*, capable of metabolizing secondary bile acids in the gut [74,75], which have a regulatory function on inflammatory pathways in the liver, colon, or white adipose tissue [76,77,78]. Our previous research noted correlated responses to selection in microbial genes involved in bile acid transport [37]; and greater plasma secondary bile acid metabolites in the L-line [36], along with more cholesterol [32], possibly linked to the greater abundance of these genera in the cecum. Also, the enrichment of *Coprococcus*, *Megasphaera*, and *Ruminoccocus* in the cecum of the L-line could relate to greater microbial propionate production [75] in line with the greater abundance of propionyl-CoA carboxylase *pccA* observed in this line [37]. Propionate inhibits de novo liver lipogenesis [79], with enzymatic activity found to be lower in the L-line [32]. It also promotes leanness by stimulating anorexigenic satiety hormones via G-protein coupled receptors, thereby inhibiting food intake, which was also lower in the L-line [80]. The L-line was also enriched in *Odoribacter*, a succinate-consuming bacteria that can prevent obesity by lowering succinate levels, known to have adipose-tissue inflammation properties [81,82]. A lower abundance of *sucC*, which catalyzes the synthesis of succinyl-CoA from succinate, was observed in the L-line in our functional metagenomics study [37]. Another interesting result was the enrichment of the ubiquitous methanogenic archaea *Methanobrevibacter* (0.52% relative abundance) in the L-line as a consequence of selection, consistent with the greater abundance of *hdr* involved in methane metabolism [37]. Obesity has been associated with increased H_2_ transfer between bacterial fermenters and H_2_-using species like methanogens [13,20,83]. Zhang et al. (2009) [84] suggested that this mechanism, which releases intraluminal pressure, promotes the fermentation of non-digestible carbohydrates and thus short-chain fatty acid production and energy intake. However, it remains unclear how increased energy production can occur while carbon is lost through methane release [85]. Moreover, acetogenesis and sulfate reduction are considered thermodynamically more favorable pathways for H_2_ depletion in monogastric animals (e.g., rabbits [86] and humans [87]) than methanogenesis. More likely, the greater methanogen abundance observed in the L-line results in increased carbon loss from the use of H_2_, reducing energy uptake, as hypothesized in pigs and humans [88,89,90]. Finally, the H-line was enriched in genera *Akkermansia*. This genus carries microbial genes involved in the biosynthesis of lipopolysaccharides, peptidoglycans, and other microbial endotoxins in its genome, which were also more abundant in the H-line [37] and are known to contribute to the development of fat mass [9,13]. Our results are contrary to expectations based on studies associating this mucin degrader with alleviating metabolic syndrome [91,92]; however, these studies primarily explored dietary interventions, not the genetic background of IMF. One hypothesis to explain our results is *Akkermansia*’s involvement in acetate production [75], which is more obesogenic than propionate. Acetate constitutes the main substrate for de novo lipogenesis in the liver of rabbits [93,94] and it also influences appetite and fat storage by promoting the secretion of insulin by the pancreas and ghrelin by the gastric mucosa [95]. 

In an attempt to transport our results into practical implementation, we identified a microbial indicator based on the abundance of four genera (*Escherichia*, *Massilistercora*, *Legionella*, and *Lysinbacillus*) that could be implemented (after an external validation) in the field of meat quality assessment to classify new animals based on their genetic background for IMF with acceptable accuracy (78%). A more sophisticated biomarker, encompassing 13 microbial genera, could increase the classification accuracy to 91%. The microbial biomarkers were built using the selbal algorithm [58], which identifies the smallest number of microbial genera that, when combined in a single microbial balance or isometric log-contrast have the highest classification accuracy. This approach differs from PLS-DA, which aims to identify all microbial variables with discriminatory ability between lines without necessarily avoiding redundancy. PLS-DA is used to comprehensively search all microbial genera that show a correlated response to selection for IMF.

## 5. Conclusions

This study highlights the impact of 10 generations of genetic selection for intramuscular fat (IMF) on the cecum microbiota of two rabbit lines maintained under identical environmental conditions but differing in genetic makeup for IMF. We identified specific microbial abundances linked to host genes responsible for lipid deposition, characterizing two divergent IMF lines. The selection exhibited a pronounced effect on microbiota composition at the genera level, with lesser effects observed at higher taxonomic ranks (phyla). The H-line was enriched in *Hungateiclostridium*, *Limosilactobacillus*, *Lysinibacillus*, *Gemminger*, and *Porphyromonas* whilst the L-line was enriched in *Megasphaera*, *Exiguobacterium*, *Flintibacter*, *Coprococcus*, *Ruminoccocus*, and *Odoribacter*. Also, the lean L-line presented a greater abundance of *Escherichia*, *Lactobacillus*, and *Methanobrevibacter*, whilst the H-line had a greater abundance of *Oxalobacter*, *Desulfovibrio*, *Bartonella*, *Legionella*, and *Akkermansia*. We designed a microbial biomarker composed of four genera abundances, which can be used to distinguish between individuals with divergent genetic backgrounds for IMF accumulation with an accuracy of up to 78% in cross-validation. These findings not only enhance our understanding of the genetic influence on microbiota composition and its role in lipid accumulation but also suggest the potential for microbiome modulation strategies to enhance IMF. This approach could open the way for future research into probiotics or microbiome-based selection aimed at optimizing meat quality.

## Figures and Tables

**Figure 1 animals-14-02078-f001:**
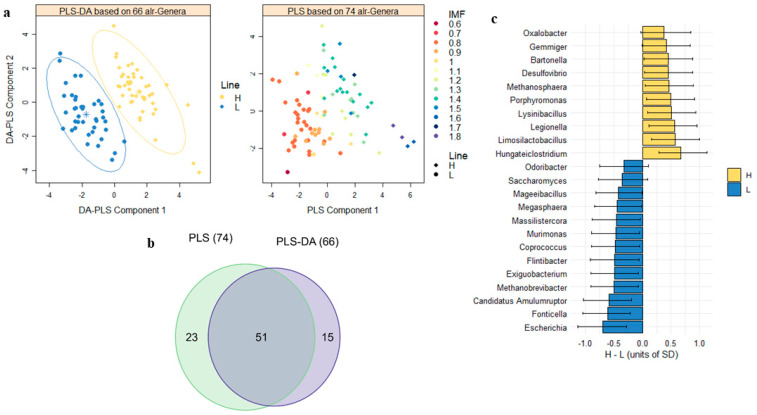
Correlated responses in microbiome composition after 10 generations of selection for intramuscular fat (IMF). (**a**). Score plot with the first two components of the partial least squares-discriminant analysis (PLS-DA) model, with 94.3% cross-validation accuracy, and of the partial least squares (PLS) model, with 65.7% cross-validation prediction accuracy. (**b**). Venn diagram shows 51 additive log-transformed (alr) genera overlapping in PLS-DA and PLS. (**c**). Differences between lines in the abundance of some of the 51 genera (means and highest posterior density intervals at 95% probability).

**Figure 2 animals-14-02078-f002:**
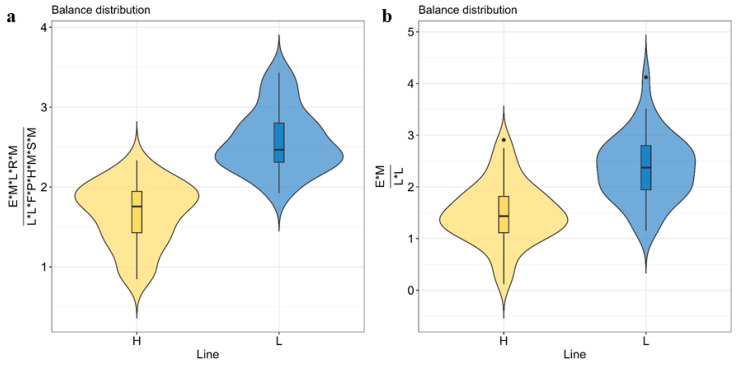
Distribution of microbial biomarkers or balances for the classification of the intramuscular fat (IMF) lines (**a**). Balance with 13 genera (geometric mean of *Escherichia*, *Megasphaera*, *Lactobacillus*, *Massilistercora*, and *Robinsoniella* in the numerator; and *Legionella*, *Lysinbacillus*, *Hungateiclostridium*, *Methanosphaera*, *Moorella*, *Faecalitalea*, *Phocaeicola*, and *Sutterella* in the denominator) (**b**). An alternative balance was built on a reduced set of four genera (using the geometric mean of *Escherichia* and *Massilistercora* in the numerator and the geometric mean of *Legionella* and *Lysinbacillus* in the denominator).

## Data Availability

Metagenomic sequence reads for all samples are available under the European Nucleotide Archive (ENA) under the accession project PRJEB46755. Resolved metagenomics (microbial gene abundances) are available upon request.

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
