# Peer review of "Correlated Responses to Selection for Intramuscular Fat on the Gut Microbiome in Rabbits"

_animals, 2024, doi:10.3390/ani14142078_

Round 1
Reviewer 1 Report
Comments and Suggestions for Authors
The article: Correlated response to selection for intramuscular fat on the gut microbiome in rabbits is interesting. However, editing is required to make it more understandable to the recipient. In the work, Materials and Methods must be well described. The same applies to the Results section.
Introduction
The description in the Introduction section generally concerns the purpose of the study and refers to the article's title. However, the reviewer suggests rewording this part to refer to the topic of the work in more detail.
Lines 74-75. "with a response of 3.8 standard deviations (SD) at the end of the experiment [26]". Is this statement valid? Does basing only on the size of the standard deviation, without the mean value, provide significant information?
Line 82. Please add a literature reference.
Are the authors sure the statement in lines 84-87 can be based on literature reports 27-29?
Line 88. "for IMF of LTL modified the IMF". I suggest avoiding only abbreviations in one short fragment.
Materials and Methods.
Did the authors specify the number of repetitions of the determinations and, therefore, the number of trials?
What about the statistical analysis of the results?
The description should be clear, so there is no need to search for individual analyses.
Results
The description of the results should refer to tables and figures.
Lines 391-398. Where are the results described by the authors presented? The same applies to other parts of the work.
Author Response
REVIEWER 1
The article: Correlated response to selection for intramuscular fat on the gut microbiome in rabbits is interesting. However, editing is required to make it more understandable to the recipient. In the work, Materials and Methods must be well described. The same applies to the Results section.
Introduction
The description in the Introduction section generally concerns the purpose of the study and refers to the article's title. However, the reviewer suggests rewording this part to refer to the topic of the work in more detail.
Lines 74-75. "with a response of 3.8 standard deviations (SD) at the end of the experiment [26]". Is this statement valid? Does basing only on the size of the standard deviation, without the mean value, provide significant information?
AUTHOR: Thank you for your comment. Indeed, breeders typically measure response to selection in units of phenotypic or genetic standard deviation due to the direct proportionality of the response achieved and the trait's variation. This approach is well-documented in numerous studies and foundational texts, such as Falconer and Mackay (1983) and Blasco (2021). In any case, we have also included the response in units of intramuscular fat (g/100g) and percentage of the mean:
Lines 74-75: with a response of 0.49 g/100g, equivalent to 46% of the mean of the trait or 3.8 standard deviations (SD), at the end of the experiment [26].
Falconer, Douglas Scott, and Trudy FC Mackay. Quantitative genetics. London: Longman, 1983.
Blasco, Agustín. "Mejora genética animal." (2021). Editorial Síntesis.
Line 82. Please add a literature reference.
AUTHOR: Reference has been added (see [27]).
Are the authors sure the statement in lines 84-87 can be based on literature reports 27-29?
AUTHOR: The referee is correct that the study of Kawai et al. [29] might not be the most appropriate to establish the affirmation. We have deleted this reference. However, the review by Fan et al. [28] is very appropriate for this statement, for example:
“Compared with the most widely used transgenic model, the mouse, rabbits have different lipoprotein metabolism features, as summarized in Table 4. For example, (1) rabbit lipoprotein profiles (low-density lipoprotein [LDL] rich) are similar to those of humans but unlike those of mice (high-density lipoprotein [HDL] rich); (2) rabbit liver does not edit apolipoprotein (apo) B mRNA and thus produces apoB-100only as does the human liver, but mouse liver also produces apoB48; therefore, apoB48 is present in both hepatically derived very low density lipoprotein (VLDL) and intesti-nally derived chylomicrons; (3) rabbits have abundant cholesteryl ester transfer protein (CETP) in their plasma as do humans whereas mice are deficient in CETP; and (4) as mentioned above, rabbits are susceptible to cholesterol-rich diet-induced atherosclerosis, whereas most strains of mice diet-induced atherosclerosis, whereas most strains of mice are resistant to cholesterol diet-induced atherosclerosis. In addition, the rabbit lacks an analogue of human apoA (hapoA)-II and has relatively lower hepatic lipase (HL) activity compared with mice and thus provides a unique system to assess the effects of these genes on plasma lipoproteins and atherosclerosis susceptibility (Brousseau & Hoeg, 1999). “
Similarly, the study by Zhao et al. [30] supports this statement; they conducted an intravenous glucose tolerance test and compared tissue adipose accumulation, observing a response akin to that in humans.
Line 88. "for IMF of LTL modified the IMF". I suggest avoiding only abbreviations in one short fragment.
AUTHOR: We have amended this as follows: … we have shown that selection for IMF of Longissimus modified the fat content.
Materials and Methods.
Did the authors specify the number of repetitions of the determinations and, therefore, the number of trials?
AUTHOR: Intramuscular fat was determined from two samples from each lyophilized muscled, each sample scanned by NIRS twice, as explained in Zomeño et al. (2011), referenced in the manuscript as [39] “… Longissimus muscle samples were scanned between 1100 and 2498 nm with a monochromator (model 5000, NIRSystem INC., Silver Spring, MD, USA) equipped with a transport module. Two round sample cups with 3.8 cm diameter quartz windows were filled with each sample and 2 spectra, rotating 90 degrees each cup, were recorded. The 4 reflectance spectra of each sample were averaged.”
The aim of the experiment is computing the correlated response to selection for IMF in the cecum microbiome, comparing the microbiome of high vs low-IMF lines. For this comparison, we took aprox. 40 replicates (n=42 form the H and 47 from the L line), as described in the manuscript.
What about the statistical analysis of the results?
AUTHOR: The paper contains a long description of the statistical analysis of the results, and all replicates were included in the analysis.
The description should be clear, so there is no need to search for individual analyses.
AUTHOR: We don’t understand what the reviewer means with “individual analysis”. In any case, we have condensed the methods section so it is easier to follow.
Results
The description of the results should refer to tables and figures.
AUTHOR: Tables and Figures are referenced along all the results section.
Lines 391-398. Where are the results described by the authors presented? The same applies to other parts of the work.
AUTHOR: While the results of microbiability are not the primary focus of the paper but rather a supplementary outcome, they are thoroughly described in the text: “Microbiability of IMF was 44.5% [15.4%, 74.8%], when computed based on microbial genera, with a probability of being greater than 25% of 0.86. When computed based on phyla abundances, it was lower, 23.7% [7.4%, 43.8%], with a probability of being greater than 25% of 0.38. We also computed microbiability based on genera calculated separately for each line, and it was still considerable, 38.8% [12.9%, 70.9%] in the H line (probability of being greater than 25% was 0.81) and 43.2% [13.5%, 76.2%] in the L line (probability of being greater than 25% was 0.83)”. Given this detailed description, we do not see the necessity for an additional table.
Reviewer 2 Report
Comments and Suggestions for Authors
The work by Martínez-Álvaro et al. is based on the hypothesis that the degree of intramuscular fat in rabbits influences the intestinal microflora and, more specifically, its composition. To verify their hypothesis, the authors used a bacterial DNA sequencing technique and tools in the form of statistical analysis including a Bayesian approach. The results are very interesting and the paper is methodologically flawless. Nevertheless, I would ask the authors to address a few issues before accepting the paper for publication.
General remark - Please rethink the notation of bacterial species. It is general rule that Italics must be used for names of phyla in text.
Line 73 - Latin names such as muscle names should be written in italics. But, why was a Latin name used at all and not an English name? Furthermore, the authors have anatomically described this muscle incorrectly. In fact, the thoracic longissimus muscle and the lumbar longissimus muscle are two separate anatomical units that are part of the 5-part longissimus muscle.
Line 131 - please explain why there is such a disproportion between males and females.
Line 142 - ad libitum
Line 151 - there is no term ‘intestinal tract’ in veterinary anatomy. It is incorrect. Please replace with ‘gastrointestinal tract’.
Figure 2 - the figure is of poor quality making it partially difficult to recognize.
Line 568 - some reference entries are in round brackets some in square brackets. This should be standardised.
Line 595 and 601 - H2 or H2
Line 620 - please remove the dot from 78%
Line 628 - the conclusion is not a summary of the work. Please rewrite this section to avoid re-describing the results obtained.
Line 644 - it seems that the conclusion formulated in this way is a way too speculative. The authors have not conducted any research in the context of obesity. This is pure speculation.
Author Response
The work by Martínez-Álvaro et al. is based on the hypothesis that the degree of intramuscular fat in rabbits influences the intestinal microflora and, more specifically, its composition. To verify their hypothesis, the authors used a bacterial DNA sequencing technique and tools in the form of statistical analysis including a Bayesian approach. The results are very interesting and the paper is methodologically flawless. Nevertheless, I would ask the authors to address a few issues before accepting the paper for publication.
General remark - Please rethink the notation of bacterial species. It is general rule that Italics must be used for names of phyla in text.
AUTHOR:Thanks for the comment, we have amended names for genus, but not for phyla. According to the International Code of Zoological Nomenclature, the usage of italics in taxonomic nomenclature typically applies to the family level and below, including species and subspecies. For taxonomic ranks above the family level, such as phyla, italics are generally not required (International Code of Zoological Nomenclature (4th Edition, 1999).
Line 73 - Latin names such as muscle names should be written in italics. But, why was a Latin name used at all and not an English name? Furthermore, the authors have anatomically described this muscle incorrectly. In fact, the thoracic longissimus muscle and the lumbar longissimus muscle are two separate anatomical units that are part of the 5-part longissimus muscle.
AUTHOR:Thanks for the comment, we have unified the manuscript using the Longissimus muscle term, without italics as it’s a common anatomical term.
Line 131 - please explain why there is such a disproportion between males and females.
AUTHOR:Rabbit lines, like other livestock genetic lines, typically consist of more females than males. This is because males can sire many offspring in a short period, allowing for stronger selection pressure via males. Thus, more females than males are selected as parents for subsequent generations. Our experiment involves 89 rabbits of 9 weeks of age, progeny of animals selected in the 9th generation, and their sexes were balanced (47 from the H line with 27 males and 20 females, and 42 from the L line with 22 males and 20 females). This detail has been clarified in the text (lines 138-140).
Line 142 - ad libitum
AUTHOR:Corrected
Line 151 - there is no term ‘intestinal tract’ in veterinary anatomy. It is incorrect. Please replace with ‘gastrointestinal tract’.
AUTHOR:Corrected
Figure 2 - the figure is of poor quality making it partially difficult to recognize.
AUTHOR:Improved. We are also including the original .TIF files in the revision as individual files.
Line 568 - some reference entries are in round brackets some in square brackets. This should be standardised.
AUTHOR:Corrected
Line 595 and 601 - H2 or H2
AUTHOR:Corrected
Line 620 - please remove the dot from 78%
AUTHOR:Corrected
Line 628 - the conclusion is not a summary of the work. Please rewrite this section to avoid re-describing the results obtained.
AUTHOR:The conclusion has been re-written accordingly and avoiding references to obesity.
Line 644 - it seems that the conclusion formulated in this way is a way too speculative. The authors have not conducted any research in the context of obesity. This is pure speculation.
AUTHOR:The conclusion has been re-written accordingly and avoiding references to obesity.
Reviewer 3 Report
Comments and Suggestions for Authors
Intramuscular fat is a key parameter to the quality of meat production. This study has investigated the correlated response to breeding for intramuscular fat on the gut microbiome in rabbits. The authors have used the bred over ten generations with high or low intramuscular fat as the models to do the analysis. The topic is very interesting. The experimental design is fine. The results demonstrate that different gut microbiome composition in hosts with divergent intramuscular fat genotypes. The findings shed the light in the field. The following revision could improve the quality of the paper.
L119, the writing of ‘2. Materials and Methods’ is too complicated. Could you please reduce it by the citation of the references?
L143, please provide the detailed diet information as a table. Please check the papers published in the Animals.
L364, The paper only have two figures, which makes the paper seems simple and do not have enough data. How about the growth performance data between the two breeds? This information will be interesting and will fulfill the perspectives of Papers.
Please do some q-PCR to very the key microbial that corelated with the intramuscular fat between the two breeders.
Please do some correlation analysis for the key different abundant of microbial between the two breeders with the intramuscular fat contents.
Please compare and discuss the findings with the similar studies in the other models, such as pigs, beef and chick etc.
If the authors can confirm the roles and mechanism of the key microbial in the formation of intramuscular fat in rabbits, the study will be more interesting.
Author Response
Intramuscular fat is a key parameter to the quality of meat production. This study has investigated the correlated response to breeding for intramuscular fat on the gut microbiome in rabbits. The authors have used the bred over ten generations with high or low intramuscular fat as the models to do the analysis. The topic is very interesting. The experimental design is fine. The results demonstrate that different gut microbiome composition in hosts with divergent intramuscular fat genotypes. The findings shed the light in the field. The following revision could improve the quality of the paper.
L119, the writing of ‘2. Materials and Methods’ is too complicated. Could you please reduce it by the citation of the references?
AUTHOR:We have condensed the methods section so it is easier to follow.
L143, please provide the detailed diet information as a table. Please check the papers published in the Animals.
AUTHOR:Done (see Table S1)
L364, The paper only have two figures, which makes the paper seems simple and do not have enough data. How about the growth performance data between the two breeds? This information will be interesting and will fulfill the perspectives of Papers.
AUTHOR:Thanks for the comment. We believe that the number of figures should correspond to the clarity and focus of the presentation, rather than the volume of data. In our case, two figures are enough to show the main results. Additionally, maintaining simplicity in the presentation is often appreciated by readers. The growth performance of the rabbits is beyond the scope of this study; including it might divert attention away from the core topics of the paper. Details about the carcass weight comparison of the lines can be found in Martínez Alvaro et al., 2016 (or [4] in the paper).
Martínez-Álvaro M, Hernández P, Blasco A. Divergent selection on intramuscular fat in rabbits: Responses to selection and genetic parameters. J Anim Sci. 2016;94: 4993–5003. doi:10.2527/jas.2016-0590
Please do some q-PCR to very the key microbial that corelated with the intramuscular fat between the two breeders.
AUTHOR:Thank you for your suggestion. We agree that this approach could significantly strengthen the findings by providing quantitative validation of the microbial taxa differentiated between the lines. Unfortunately, due to current limitations in our laboratory resources and budget, we are unable to carry out q-PCR assays at this stage. Our facility does not have the necessary equipment, and the funding allocated for this project does not cover the additional costs that these procedures would entail.
Please do some correlation analysis for the key different abundant of microbial between the two breeders with the intramuscular fat contents.
AUTHOR:Thanks for the suggestion. A correlation analysis between IMF and the 51 microbial genera is found below. However, we have not included this in the manuscript, because the highlighted microbial genera are related to genetic values of IMF, and not the phenotypic value of IMF. Thus, we think it could confuse the reader.
|
Table. Person correlation between the 51 microbial genera alr-abundances (corrected by sequencer and sex effects) and IMF |
||
|
|
Pearson correlation |
s.e. |
|
Ruminococcus |
-0.23 |
0.10 |
|
Akkermansia |
0.19 |
0.10 |
|
Marvinbryantia |
-0.16 |
0.10 |
|
Oxalobacter |
0.23 |
0.10 |
|
Odoribacter |
-0.10 |
0.10 |
|
Phocaeicola |
0.15 |
0.10 |
|
Saccharomyces |
-0.27 |
0.10 |
|
Coprococcus |
-0.27 |
0.10 |
|
Methanobrevibacter |
-0.18 |
0.10 |
|
Candidatus Amulumruptor |
-0.18 |
0.10 |
|
Lactobacillus |
-0.19 |
0.10 |
|
Desulfovibrio |
0.28 |
0.10 |
|
Gemmiger |
0.25 |
0.10 |
|
Robinsoniella |
-0.18 |
0.10 |
|
Murimonas |
-0.16 |
0.10 |
|
Flintibacter |
-0.24 |
0.10 |
|
Megasphaera |
-0.15 |
0.10 |
|
Listeria |
0.17 |
0.10 |
|
Desulfitobacterium |
0.15 |
0.10 |
|
Faecalicatena |
0.17 |
0.10 |
|
Methanosphaera |
0.18 |
0.10 |
|
Lactonifactor |
0.16 |
0.10 |
|
Azospirillum |
0.17 |
0.10 |
|
Escherichia |
-0.33 |
0.09 |
|
Actinomyces |
-0.13 |
0.10 |
|
Hungateiclostridium |
0.28 |
0.10 |
|
Sutterella |
0.22 |
0.10 |
|
Kazachstania |
-0.20 |
0.10 |
|
Trichuris |
0.21 |
0.10 |
|
Moorella |
0.17 |
0.10 |
|
Massilistercora |
-0.25 |
0.10 |
|
Porphyromonas |
0.30 |
0.10 |
|
Lachnobacterium |
0.14 |
0.10 |
|
Limosilactobacillus |
0.18 |
0.10 |
|
Catonella |
0.17 |
0.10 |
|
Catenibacterium |
-0.20 |
0.10 |
|
Marasmitruncus |
0.18 |
0.10 |
|
Ndongobacter |
-0.12 |
0.10 |
|
Erysipelatoclostridium |
0.24 |
0.10 |
|
Microbacterium |
-0.18 |
0.10 |
|
Turicibacter |
-0.35 |
0.09 |
|
Mageeibacillus |
-0.12 |
0.10 |
|
Exiguobacterium |
-0.10 |
0.10 |
|
Aminipila |
0.21 |
0.10 |
|
Lysinibacillus |
0.24 |
0.10 |
|
Paraprevotella |
0.21 |
0.10 |
|
Bartonella |
0.22 |
0.10 |
|
Faecalitalea |
0.21 |
0.10 |
|
Legionella |
0.32 |
0.09 |
|
Fonticella |
-0.30 |
0.10 |
|
Massilicoli |
0.18 |
0.10 |
Please compare and discuss the findings with the similar studies in the other models, such as pigs, beef and chick etc.
AUTHOR:Only Sapp et al. (2002), in cattle; Zhao et al. (2007), in chickens; and Schwab et al. (2009), in pigs, performed selection for IMF. To our knowledge, they did not study the microbiome of their genetic lines. Please note that our experimental design allows us to identify microbial abundances influenced by specific genes for IMF. This is distinct from comparing groups with varying IMF levels, which could involve multiple confounding factors such as different breeds, genetic backgrounds, and diets. However, in the discussion section, we do compare our results with other studies in livestock species, as detailed in lines 486-487:
: “Although there are no previous records of IMF microbiability in rabbits, other studies performed in pigs provided lower values ranging from 6% [63] to 26% [64,65].”
Additionally, we reference studies that associated our identified taxa, such as Gemminger, Porphyromonas, and methanogens, with fat deposition in other species. For instance, Luo et al. [90] noted:
“In literature, Gemminger was found to be more abundant in obese children than in normal weight individuals [71]; Erysipelatoclostridium, was linked to diet-induced obesity in gnotobiotic mice [72]; and periodontal infection with Porphyromonas induced obesity, glucose tolerance, hepatic steatosis and altered endocrine function in mice [73,74].”
“More likely, the greater methanogens abundance observed in the L-line is hypothesized to result in increased carbon loss from the use of H2, reducing energy uptake, as hypothesized in pigs and humans [90–92]”.
Schwab, C. R., T. J. Baas, K. J. Stalder, and D. Nettleton. 2009. Results from six generations of selection for intramuscular fat in Duroc swine using real-time ultrasound. I. Direct and correlated phenotypic responses to selection. J. Anim. Sci. 87:2774–2780. doi:10.2527/jas.2008-1335
Sapp, R. L., J. K. Bertrand, T. D. Pringle, and D. E. Wilson. 2002. Effects of selection for ultrasound intramuscular fat percent- age in Angus bulls on carcass traits of progeny. J. Anim. Sci. 80:2017–2022. doi:10.2527/2002.8082017x
Zhao, G. P., J. L. Chen, M. Q. Zheng, J. Wen, and Y. Zhang. 2007. Correlated responses to selection for increased intramuscular fat in a Chinese quality chicken line. Poult. Sci. 86:2309–2314. doi:10.1093/ps/86.11.2309
If the authors can confirm the roles and mechanism of the key microbial in the formation of intramuscular fat in rabbits, the study will be more interesting.
AUTHOR:Thank you for your comment. In our previous study (Martínez-Álvaro et al., 2021) [37], we compared the functional metagenomics across H and L lines, which is why we frequently reference this work in the discussion. While additional omics analyses such as meta-transcriptomics or meta-proteomics could indeed make our findings more interesting, we have not yet conducted these studies. Using gnotobiotic animal models to study the effect of specific microbes on intramuscular fat deposition would be also a promising direction for future research. However, our efforts are currently constrained by our limited laboratory resources and limited funding.
Round 2
Reviewer 1 Report
Comments and Suggestions for Authors
Manuscript was improved.
The reviewer thanks the authors for their response.
Author Response
Many thanks!
Reviewer 3 Report
Comments and Suggestions for Authors
It is very upset that the authors did not followed the suggestion to improve the quality of the paper.
The ‘2. Materials and Methods’ still could be reduced.
The data was too less for a paper.
Add the growth performance data between the two breeds?
Please do some q-PCR to very the key microbial that corelated with the intramuscular fat between the two breeders.
Please do some correlation analysis for the key different abundant of microbial between the two breeders with the intramuscular fat contents.
Please compare and discuss the findings with the similar studies in the other models, such as pigs, beef and chick etc. Summarized these ifnormation as the data.
Author Response
REFEREE 3: “It is very upset that the authors did not followed the suggestion to improve the quality of the paper”.
AUTHORS: The authors can disagree with referees, and improve the quality of the paper by no following some suggestions, as far as it is duly justified.
REFEREE 3: “The ‘2. Materials and Methods’ still could be reduced”.
AUTHORS: Nowadays it is disgraceful that M&M section takes less and less importance, because often readers cannot understand well how the experiment was conducted. We do not find necessary to remind that M&M should be developed with detail enough to permit the reader to repeat the experiment.
REFEREE 3: “The data was too less for a paper”.
AUTHORS: Our sample size is greater than other sample sizes in similar papers published in Animals, like Kim et al. 2020 (n=14), Lei et al. 2021 (n=16) or Owens et al. (n=13), and we have a unique experimental design which allows us to reach conclusions about genetics with a reduced sample size. As the lines were contemporarily reared in the same environment (same building, same food, same management, etc.), their differences are exclusively due to genetic causes underlying IMF.
REFEREE 3: “Add the growth performance data between the two breeds?”
AUTHORS: We did not include growth because lines were selected for a different trait (IMF) almost uncorrelated to growth; we cannot include a list of uncorrelated phenotypes in the paper because it would be of little utility for the discussion and the conclusions.
REFEREE 3: “Please do some q-PCR to very the key microbial that correlated with the intramuscular fat between the two breeders”.
AUTHORS: As explained before, due to current limitations in our laboratory resources and budget, we are unable to carry out q-PCR assays at this stage.